# Higher HEI-2015 Score Is Associated with Reduced Risk of Depression: Result from NHANES 2005–2016

**DOI:** 10.3390/nu13020348

**Published:** 2021-01-25

**Authors:** Kai Wang, Yudi Zhao, Jiaqi Nie, Haoling Xu, Chuanhua Yu, Suqing Wang

**Affiliations:** 1Department of Nutrition and Food Hygiene, School of Health Science, Wuhan University, 185, Donghu Rd, Wuhan 430071, China; wk1148761305@whu.edu.cn (K.W.); 15502730223@163.com (J.N.); xuhlntl@163.com (H.X.); 2Department of Epidemiology and Biostatistics, School of Health Sciences, Wuhan University, 185 Donghu Rd, Wuhan 430071, China; yudi_zhao@whu.edu.cn (Y.Z.); yuchua@whu.edu.cn (C.Y.)

**Keywords:** healthy eating index, depression, NHANES, diet pattern, DGA

## Abstract

Globally, the total estimated number of people living with depression increased by 18.4% between 2005 and 2015, with the prevalence being 4.8% in 2015. Many nutrient and diet patterns are proven to be correlated to depression, so we conducted this analysis to explore whether the Healthy Eating Index 2015 (HEI-2015) score is associated with depression, and possibly to provide dietary measures to reduce the risk of depression. Data came from the National Health and Nutrition Examination Survey (2005–2016), a cross-sectional and nationally representative database. The analytic sample was limited to adults: (1) age ≥20 with complete information of HEI-2015 and depression; (2) no missing data of demographics, BMI, drinking, smoking, and fasting plasma glucose. HEI-2015 was calculated using the Dietary Interview: Total Nutrient Intakes, First Day data file. Depression was assessed using the Patient Health Questionnaire-9 (PHQ-9). Weighted logistic regression models were used to explore the relationship between the HEI-2015 score and depression. The final study sample included 10,349 adults, with 51.4% of them being men, representing a population of about 167.8 million non-institutionalized U.S. adults. After multivariable adjustment, average HEI status (OR: 0.848, 95% CI: 0.846–0.849) and optimal HEI status (OR: 0.455, 95% CI: 0.453–0.456) were associated with reduced odds of depression. Poor diet quality is significantly associated with elevated depressive symptoms in U.S. adults. Aligning with the Dietary Guidelines for Americans reduces the risk of depression.

## 1. Introduction

Depression, clinically characterized by significant and persistent low mood symptoms, is a common and growing globally mental health issue linked with considerably diminished role-functioning and quality of life, medical comorbidity, and mortality [1]. In 2017, about 17.3 million adults aged 18 and over in the US had experienced at least one major depressive episode. The prevalence was about 7.1%, and highest among adults reporting two or more races [2]. According to the Depression and Other Common Mental Disorders: Global Health Estimates published by WHO in 2017, the total number of people living with depression in the world is 322 million. Nearly half of them live in the South-East Asia Region and Western Pacific Region, such as China and India. Meanwhile, the prevalence of depression varies by age, peaking in older adulthood, and was estimated above 7.5% among females aged 55–74 years [3]. A substantial number of researches have shown strong relationships between depression and physical health, including cardiovascular disease [4], Parkinson’s disease [5], metabolic disease [6], dementia [7], type 2 diabetes [8], and cancer [9]. Out of the mental and addictive disorders, depressive disorders cause most disability-adjusted life years for both sexes, followed by anxiety disorders in women [10]. In 2015, depressive disorders led to a global total of over 50 million years lived with disability (YLD), more than 80% of which occurred in low- and middle-income countries [4]. Studies in recent decades have shown associations between nutrient intake and the risk of depression, including minerals like zinc, omega-3 fatty acids, and vitamins such as vitamin D [11,12,13,14]. Many researchers also found adherence to a specific dietary pattern, such as “dietary approaches to stop hypertension”, was correlated with lower depression risk [15,16,17]. While most of them focused on specific diet pattern or food intake, some research explored the relationship from a macroscopic view. To explore the effect of diet quality in a more macroscopic way, we adopted the latest edition of the Healthy Eating Index (HEI) to determine whether diet quality is related to depression. Through this research, we want to answer whether aligning with the Dietary Guidelines for Americans (DGA) reduces the risk of depression.

## 2. Materials and Methods

### 2.1. Sample

Data came from the six continuous National Health and Nutrition Examination Survey (NHANES) cycles from 2005–2016 (https://wwwn.cdc.gov/nchs/nhanes/ContinuousNhanes/Default.aspx?BeginYear=2005). NHANES is a nationally representative, population-based survey for assessing adult and child health and nutritional status in the US. This survey combined health interviews conducted in respondents’ homes with health measurements (e.g., DPQ_I, objective physical measures) performed at mobile exam centers (MECs). The examination components consisted of medical, dental, and physiological measurements, and laboratory tests supervised by trained medical personnel. Furthermore, the adoption of various modern equipment enabled the NHANES to collect reliable, high-quality data. Moreover, compensation and a report of medical findings were given to each participant, which increased the compliance of participants [18]. The total sample size of adults from the 2005–2016 assessments was N = 10,349. Additional details of the study design, sampling, and exclusion criteria are described in Figure 1. Only publicly available data was used in the analysis, and no ethical approval was needed in this study.

### 2.2. Measures

Diet quality: The Healthy Eating Index (HEI) is a measure for assessing dietary quality, precisely, the degree to which a set of foods aligns with the Dietary Guidelines for Americans [19]. The HEI-2015 components were the same as in the HEI-2010, except saturated fat and added sugars replaced empty calories, with the result being 13 components [20]. HEI-2015 scores ranged from 0–100, with higher HEI scores reflecting better diet quality. We utilized the total nutrient intakes on the first day (DR1TOT) to calculate the 13 components of HEI-2015. For further weighted Scott–Rao chi-square test and weighted logistic regressions, an HEI-2015 score less than 50, between 50 and 70, and more than 70 was categorized as inadequate, average and optimal, respectively [21].

Depression: Current depressive symptoms were measured by the Patient Health Questionnaire-9 (PHQ-9). The PHQ-9 is a well-validated (Cronbach’s α = 0.89) self-report instrument that assesses depression symptoms (i.e., sadness, trouble sleeping, fatigue, problems concentrating) in the past two weeks, and has moderate concordance with clinical psychiatric interviews. The PHQ-9 questionnaire contains nine items, with each item being assessed on a four-point Likert scale, ranging from 0 = not at all to 3 = nearly every day, and summing up a total scale range of 0 to 27. A dichotomous variable indicating no depression (PHQ-9 score <10) or elevated depressive symptoms (PHQ-9 score ≥10) was created using a threshold score of 10 [22].

#### Covariates

Each covariant was categorized into a reference group, and other groups. When analyzing, all other groups were compared to this reference group to estimate the relative odds ratio.

Sex: Sex was categorized as male (reference group) and female.

Age: Age was categorized as 20 to 25 years (reference group), 26 to 49 years, and 50+ years [4].

Race: Race was categorized as non-Hispanic white (reference group), non-Hispanic black, Mexican American, and other races [23].

Education level: Education level was categorized as less than a high school diploma (reference group), high school graduate/GED, some college/AA degree, and college graduate or more [24].

Household income: Household income was categorized as ≤130% (reference group), >130% to 350%, and >350% by the ratio of family income to poverty (FPL) [25].

BMI status: body mass index was calculated from measured height and weight as weight/height^2^ (kg/m^2^), and then categorized into ≤25 (reference group), >25 to 30, and >30 [26].

Smoking status: smoking behavior was measured in the “smoking: cigarette use” questionnaire. In the “smoking: cigarette use” questionnaire, respondents were asked if s/he had smoked at least 100 cigarettes in their life, and smoked cigarettes when being questioned. If the respondent had smoked less than 100 cigarettes in their life, s/he was classified as a never smoker. If the respondent had smoked at least 100 cigarettes in his/her life and still smoked when s/he answered the questionnaire, s/he was classified as a current smoker. The respondent was classified as a former smoker if s/he had smoked at least 100 cigarettes in his/her life, and had quit smoking when s/he answered the questionnaire. Smoking status was categorized into never smoker (reference group), former smoker, and current smoker [27].

Drinking status: drinking behavior was measured in the “alcohol use” questionnaire. In the “alcohol use” questionnaire, each respondent was asked how often s/he had drunk alcoholic drinks in the past 12 months, and the average drinks on those days that s/he drank alcoholic beverages. According to these questions, the average number of alcoholic drinks consumed per week in the past 12 months could be calculated. Then it was categorized into four strata (0, <1, 1–<8, and ≥8 drinks per week) and defined as none (reference group), light, moderate, and heavy alcohol consumption, respectively. A “drink” was defined as a 12-ounce beer, a 5-ounce glass of wine, or one-and-half ounces of liquor [28].

Diabetes: plasma glucose data were obtained from the plasma fasting glucose laboratory data. Respondents whose fasting plasma glucose was ≥6.0 mmol/L were thought to be a diabetic, consistent with American Diabetes Association guidelines [29]. Thus, respondents were categorized into adults with normoglycemia (reference group), and adults with diabetes.

### 2.3. Statistical Analysis

Initially, the trends of depression and other characteristics in the six continuous cycles were estimated with the Cochran–Armitage trend test. Then, the baseline characteristics of different groups were compared using the weighted Scott–Rao chi-square test [30]. HEI scores were described with a median (P25, P75). Finally, a series of weighted steps forward (likelihood ratio) binary logistic regression models were fit to assess the relationship between diet quality and depression. Estimates were weighted to be representative of the general adult population. All *p* values refer to two-tailed tests. Statistical analyses were conducted using the SPSS statistical package (Version 23.0; SPSS Inc., Chicago, IL, USA).

## 3. Results

Figure 1 described the study design, sampling, and exclusion; and 18,006 participants were excluded because of missing data on any of the covariates. Among them, 2361, 12, 219, 12, 3594, and 11,808 individuals were excluded because of missing data on income, education, BMI, smoking, drinking, and diabetes, respectively. Our final sample included 10,349 NHANES participants, representing a population of about 167.8 million non-institutionalized U.S. adults, with 48.6% being female and 72.0% being non-Hispanic White.

Table 1 described the prevalence of depression and associated characteristics in six continuous NHANES cycles, in which the trend tests were also conducted. The prevalence of depression grew with time, from 4.8% in 2005–2006 to 7.4% in 2015–2016. In addition, there are other points worth noting. For example, the proportion of women and adults aged over fifty years old increased with time. The proportion of adults with normal or low weight decreased with time, indicating the urgency of body shape management. The prevalence of diabetes increased with time, which reminds adults of the significance of blood glucose control. The proportion of adults with inadequate HEI status decreased with time, showing the improving diet quality in the six cycles.

Table 2 describes the characteristics of participants with the weighted Scott–Rao chi-square test. Adults with depression were more likely to be female, non-Hispanic Black, obese, over 50 years old, current smokers, diabetic, alcoholic, have less than high school education, have low household income, and have inadequate HEI scores.

Table 3 shows the results of three weighted logistic regression models. Model 1 was adjusted for demographics characteristics (i.e., sex, age group, race, income, and education). Model 2 was adjusted for all Model 1 covariates and BMI, smoking, and drinking status. Moreover, Model 3 was adjusted for all Model 2 covariates and diabetes. After adjusting for demographic characteristics, optimal HEI status was associated with 0.378 times lower odds (95% CI, 0.377–0.379) of current depression, relative to inadequate HEI status. Additional adjustment for BMI, smoking, drinking, and diabetes status did not substantially attenuate these relationships. After multivariable adjustment, adults with average HEI status (OR: 0.848, 95% CI: 0.846–0.849) and optimal HEI status (OR: 0.455, 95% CI: 0.453–0.456) were associated with reduced odds of depression. Adults with diabetes were more likely to suffer from depression, with the odds ratio being 1.637 (95% CI: 1.634–1.640).

Figure 2 shows the trend of the HEI-2015 score in the form of a violin plot. HEI score increased from 47.77 (39.39, 56.51) in 2005–2006, to 50.74 (43.13, 59.13) in 2015–2016. The proportion of adults with inadequate HEI status decreased with time, and that of adults with optimal HEI status increased with time.

Figure 3 shows the results of three weighted logistic regression models in the form of a forest plot. As is shown, average and optimal HEI status are both protective factors for depression, reducing the depression risk by 15.2% and 54.5%. Diabetes is the risk factor of depression, increasing the depression risk by 63.7%. Cigarette smoking and heavy drinking are both behavioral risk factors.

## 4. Discussion

The results revealed that a higher HEI score was significantly correlated to less elevated depression symptoms, so we concluded that higher diet quality was significantly correlated to a lower risk of depression. Our findings also suggest that Mexican Americans are less likely to suffer from depression, which needs further analyses of genetic factors.

We analyzed the depression status of participants from nine symptomatic questions from the PHQ in 2005–2016 NHANES data, and the results revealed a depression prevalence of 6.9%. Since the sample is generalizable to the non-institutionalized civilian U.S. population, we assume the prevalence to be credible, similar to the data published by the WHO in 2017.

To our knowledge, many factors are associated with the occurrence of depression, for instance, alcohol consumption and diabetes. Moreover, it has been found by many researchers that women are more susceptible to depression than men [31,32,33], in accordance with our results (odds ratio, 1.889). Moreover, three logistic regression models were adopted in this analysis to explore a more appropriate model.

Some results have been widely recognized by many researchers. For example, the results of the weighted logistic models revealed that the risk of adults aged over 50 years old suffering from depression was 1.827 times that of adults aged 20–24 years old. The report from the WHO also concluded that the prevalence varies by age, peaking in older adulthood, similar to our results [3]. After multivariable adjustment, the odds ratio for other races changed from less than one, to more than one, which needs more specific classification. Our results revealed higher education level reduced the depression risk, similar to other studies [34,35,36]. In addition, our results revealed a negative correlation between household income and depression, consistent with other research [37,38,39]. It is recognized by many experts that a positive association exists between smoking and depression, as in our results [40,41,42]. Among the results, we found some interesting facts. Compared with normal and low weight participants, overweight adults were less likely to suffer from depression, contrary to our original thoughts. However, we found that Z Ul-Haq got a similar result from a cross-sectional study consisting of 37,272 participants, which revealed that overweight participants had better mental health than the normal-weight group [43]. Generally, overweight and obese adults suffer more ridicule and gossip than normal and low weight adults, which may be a reason for depression. After further analyses, we found that overweight adults accounted for about one third of American adults, so were obese adults. This meant high BMI is typical among American adults, and discrimination in the US is not as high as in China, leading to less psychological pressure in overweight adults. At the same time, overweight adults relieve pressure through diet, and are less susceptible to depression.

Another fact is that light and moderate drinking is a protective factor for depression. A meta-analysis in 2013 concluded that light drinking increased the risk of cancer of the oral cavity and pharynx, esophagus, and female breast [44]. However, many researchers have found that light and moderate drinking also have some health benefits, including reducing the risk of dementia [45], heart failure [46], ischemic stroke [47], type 2 diabetes [48], and all-cause mortality [49]. In a cohort study with ten years of follow up, a J-shaped association was found with increased psychological distress among abstainers and heavy drinkers compared to light or moderate drinkers [50]. Our analysis believes that light and moderate drinking helps people deal with emotional issues and refresh themselves, thus reducing the risk of depression.

Physical activity has been found to be correlated to depression in a substantial number of studies [51,52,53,54]. However, three versions of the physical activity questionnaire were adopted in 2005–2016, and no identical and detailed information could be used to conduct an analysis. Therefore, physical activity was not included as a covariant in this analysis.

Diet quality was reported to be correlated to diabetes in many studies, not only in randomized controlled trials, but also in large population-based cohorts [55,56]. The relationship between diabetes and depression was recognized by R J Anderson in 2001 [57], yet whether there is a causal relationship is still under debate.

Thus, we conducted a mediation analysis to figure out whether diet quality influenced depression through diabetes. The HEI-2015 score was the independent variable, and the PHQ-9 score was the dependent variable. The result showed a mediating effect, but the mediating effect’s proportion was only 0.14%. For further mechanism analysis, we need to explore more possible factors causing the mediation effect.

The present study has several limitations. The cross-sectional design of the study is the primary limitation, and no causation should be inferred from this study. Second, using self-reported 24-h dietary recall data and the PHQ-9 questionnaire is a limitation, as they are subject to over- or under-reporting. Finally, 18,006 participants were excluded because of missing data on any of the covariates, which may have affected the results. Despite these limitations, our study has several strengths. Using a large, nationally representative database to estimate diet quality is a major strength of the present study. Adopting the latest edition of HEI is another strength. Moreover, data in six cycles were combined to increase the sample size.

Since we found that a higher HEI-2015 score is associated with a lower risk of depression, the next step of our plan is to figure out the pathway by which HEI-2015 influences depression, with structural equation models.

By virtue of this article, we would like to make the public aware of the significance of better diet quality on depression. Since better diet quality is associated with less depression risk, why do we not improve our diet quality to reduce the risk of depression?

## 5. Conclusions

This study’s primary finding is that depression is rapidly growing in prevalence among American adults, from 4.8% in 2005–2006, to 7.4% in 2015–2016. Poor diet quality is significantly associated with elevated depressive symptoms. An optimal HEI-2015 score reduces the risk of suffering from elevated depressive symptoms by 54.5% compared to an inadequate HEI-2015 score.

## Figures and Tables

**Figure 1 nutrients-13-00348-f001:**
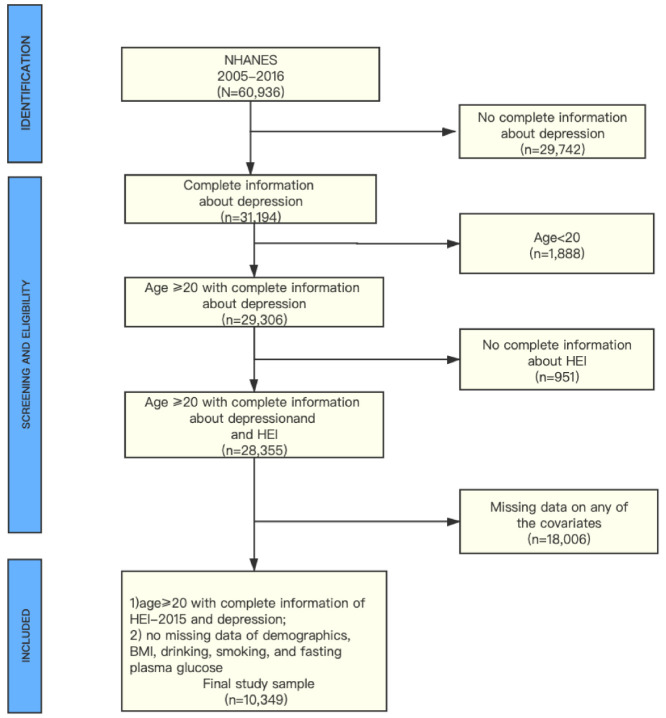
Flowchart of the study population.

**Figure 2 nutrients-13-00348-f002:**
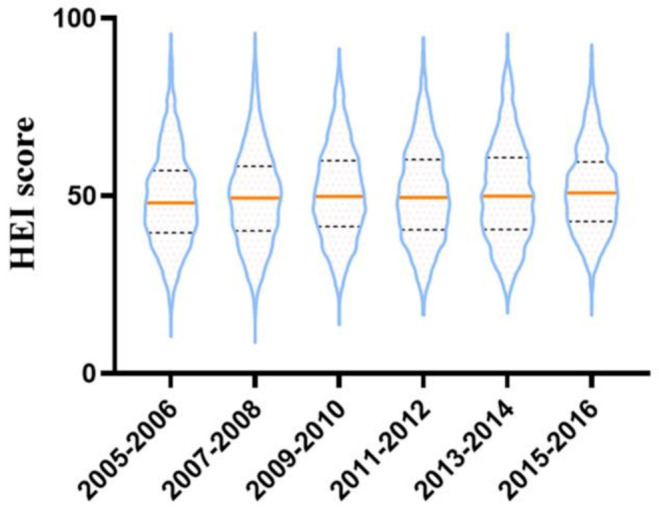
The trends of HEI-2015 score in the six cycles from 2005–2016 in NHANES.

**Figure 3 nutrients-13-00348-f003:**
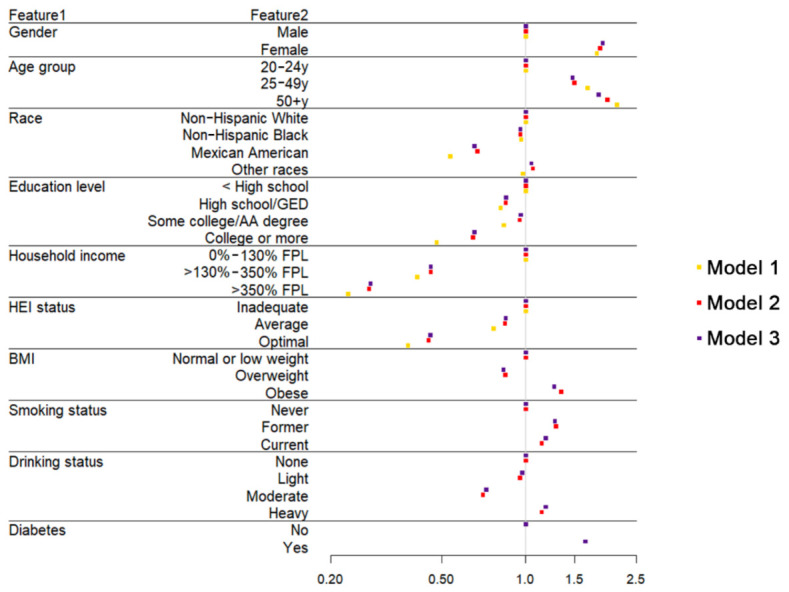
The forest plot shows the odds ratios of analyzing variables in three weighted logistic regression models.

**Table 1 nutrients-13-00348-t001:** US trends in characteristics among adults aged 20 years or older.

Characteristics	2005–2006 (n = 1595)	2007–2008 (n = 1839)	2009–2010 (n = 1913)	2011–2012 (n = 1619)	2013–2014 (n = 1778)	2015–2016 (n = 1605)	*p* Trend
No. with depressive symptoms	84	157	166	131	159	139	
The prevalence of depression (Weighted %)	4.8	6.8	7.0	7.4	8.0	7.4	<0.001
**Sex (No. Weighted %)**							<0.001
Male	856 (51.2)	993 (52.7)	983 (51.4)	891 (52.0)	710 (50.7)	856 (50.6)	
Female	739 (48.8)	846 (47.3)	930 (48.7)	728 (48.0)	868 (49.3)	749 (49.4)	
**Age group (No. Weighted %)**							<0.001
20–24 y	151 (8.9)	136 (9.1)	175 (9.3)	171 (9.6)	143 (9.5)	115 (7.2)	
25–49 y	746 (51.5)	775 (51.0)	809 (47.6)	709 (46.3)	764 (45.4)	670 (44.8)	
50+ y	698 (39.6)	928 (39.9)	929 (43.1)	739 (44.1)	871 (45.0)	820 (45.8)	
**Race (No. Weighted %)**							<0.001
Non-Hispanic White	865 (75.2)	932 (72.6)	1023 (73.9)	728 (71.9)	868 (70.2)	646 (68.7)	
Non-Hispanic Black	334 (9.7)	364 (10.4)	298 (9.8)	366 (10.1)	339 (10.7)	321 (10.1)	
Mexican American	281 (7.2)	304 (8.0)	331 (7.3)	159 (6.6)	224 (8.3)	240 (7.1)	
Other races	115 (7.9)	239 (9.0)	261 (9.0)	366 (11.4)	347 (10.8)	398 (14.1)	
**Education level (No. Weighted %)**							<0.001
<High school	373 (14.8)	509 (17.5)	475 (16.2)	323 (14.0)	345 (14.2)	306 (11.9)	
High school/GED	400 (26.2)	453 (23.8)	434 (22.7)	340 (18.7)	384 (20.1)	374 (22.3)	
Some college/AA degree	483 (32.8)	478 (29.4)	567 (30.6)	500 (33.1)	557 (33.2)	506 (32.6)	
College or more	339 (26.1)	399 (29.4)	437 (31.4)	456 (34.2)	492 (32.5)	419 (33.1)	
**Household income (No. Weighted %)**							<0.001
0–130% FPL	352 (13.3)	503 (17.3)	598 (20.5)	540 (21.8)	574 (23.1)	463 (18.2)	
>130–350% FPL	652 (39.1)	718 (33.6)	711 (36.0)	576 (36.6)	610 (34.3)	665 (37.9)	
>350% FPL	591 (47.6)	618 (49.1)	604 (43.4)	503 (41.9)	594 (42.6)	477 (43.3)	
**BMI status (No. Weighted %)**							<0.001
Normal or low weight	479 (32.0)	538 (31.6)	540 (31.0)	503 (31.2)	535 (30.0)	422 (26.1)	
Overweight	537 (31.9)	653 (36.1)	665 (34.0)	528 (33.6)	589 (33.2)	534 (32.5)	
Obese	579 (36.2)	648 (32.3)	708 (34.9)	588 (35.2)	654 (36.9)	649 (41.4)	
**Smoking status (No. Weighted %)**							<0.001
Never	721 (44.6)	863 (50.1)	950 (51.4)	853 (54.1)	911 (53.2)	767 (48.1)	
Former	482 (28.5)	528 (26.7)	531 (28.0)	413 (25.5)	481 (26.8)	469 (31.1)	
Current	392 (27.2)	448 (23.1)	432 (20.6)	353 (20.5)	386 (20.0)	369 (20.8)	
**Drinking status (No, Weighted %)**							<0.001
None	394 (19.0)	434 (18.5)	385 (16.1)	315 (15.3)	362 (15.6)	307 (15.7)	
Light	516 (33.4)	620 (33.9)	682 (34.0)	591 (34.9)	656 (36.5)	606 (36.0)	
Moderate	457 (30.7)	514 (32.1)	546 (33.3)	453 (31.2)	522 (33.3)	483 (34.3)	
Heavy	228 (16.8)	271 (15.5)	300 (16.7)	238 (18.6)	238 (14.6)	209 (13.9)	
**Diabetes (No. Weighted %)**							<0.001
No	1448 (93.0)	1617 (92.6)	1707 (92.3)	1445 (92.5)	1570 (90.2)	1370 (88.7)	
Yes	147 (7.0)	222 (7.4)	206 (7.7)	174 (7.5)	208 (9.8)	235 (11.3)	
**HEI status (No. Weighted %)**							<0.001
Inadequate	889 (57.2)	959 (55.2)	958 (49.3)	825 (49.7)	895 (50.9)	758 (47.3)	
Average	594 (36.4)	751 (38.6)	789 (41.3)	652 (39.8)	704 (40.1)	720 (45.1)	
Optimal	112 (6.4)	129 (6.1)	166 (9.4)	142 (10.5)	179 (9.1)	127 (7.5)	

Values are survey-weighted percentages. FPL = family income to poverty. HEI = healthy eating index.

**Table 2 nutrients-13-00348-t002:** Characteristics among adults aged 20 years or older by depression.

Characteristics	Adults without Depression	Adults with Depression	*p*-Value
No. (Weighted %)	9513 (93.1)	836 (6.9)	
**Sex**			<0.001
Male	5159 (52.4)	330 (38.1)	
Female	4354 (47.6)	506 (61.9)	
**Age group**			<0.001
20–24 y	4129 (9.5)	326 (7.4)	
25–49 y	3212 (47.9)	368 (45.6)	
50+ y	2172 (42.6)	142 (47.1)	
**Race**			<0.001
Non-Hispanic White	4654 (72.3)	408 (69.1)	
Non-Hispanic Black	1846 (9.9)	176 (13.5)	
Mexican American	1430 (7.5)	109 (6.2)	
Other races	1583 (10.3)	143 (11.2)	
**Education level**			<0.001
<High school	2052 (14.0)	279 (24.9)	
High school/GED	2176 (21.8)	209 (26.3)	
Some college/AA degree	2841 (31.8)	250 (34.4)	
College or more	2444 (32.4)	98 (14.5)	
**Household income**			<0.001
0–130% FPL	2574 (17.4)	456 (42.3)	
>130–350% FPL	3667 (36.2)	265 (35.7)	
>350% FPL	3272 (46.4)	115 (22.0)	
**BMI status**			<0.001
Normal or low weight	2795 (30.4)	222 (28.7)	
Overweight	3293 (34.2)	213 (24.8)	
Obese	3425 (35.4)	401 (46.5)	
**Smoking status**			<0.001
Never	4787 (51.6)	278 (31.8)	
Former	2695 (28.0)	209 (24.3)	
Current	2031 (20.3)	349 (44.0)	
**Drinking status**			<0.001
None	1979 (16.2)	218 (23.0)	
Light	3336 (34.6)	335 (37.9)	
Moderate	2833 (33.4)	142 (20.6)	
Heavy	1365 (15.8)	141 (18.5)	
**Diabetes**			<0.001
No	8463 (92.0)	694 (85.6)	
YES	1050 (8.0)	142 (14.4)	
**HEI status**			<0.001
Inadequate	4783 (50.8)	501 (62.4)	
Average	3908 (40.7)	302 (34.3)	
Optimal	822 (8.5)	33 (3.2)	

Values are survey-weighted percentages. FPL = family income to poverty.

**Table 3 nutrients-13-00348-t003:** Relationship between HEI and Depression among Adults aged 20 years or older.

Variable	OR (95% CI)		
	Model 1	Model 2	Model 3
**Sex (reference, Male)**			
Female	1.799 (1.787, 1.791)	1.850 (1.847, 1.852)	1.889 (1.887, 1.892)
**Age group (reference, 20–24 y)**			
25–49 y	1.669 (1.665, 1.673)	1.495 (1.491, 1.499)	1.472 (1.468, 1.476)
50+ y	2.128 (2.123, 2.133)	1.968 (1.963, 1.973)	1.827 (1.822, 1.832)
**Race (reference, Non-Hispanic White)**			
Non-Hispanic Black	0.963 (0.961, 0.965)	0.958 (0.956, 0.960)	0.958 (0.956, 0.960)
Mexican American	0.535 (0.534, 0.537)	0.670 (0.668, 0.672)	0.656 (0.655, 0.658)
Other races	0.976 (0.974, 0.978)	1.062 (1.060, 1.064)	1.048 (1.046, 1.050)
**Education level (reference, <High school)**			
High school/GED	0.812 (0.810, 0.813)	0.848 (0.846, 0.849)	0.852 (0.850, 0.853)
Some college/AA degree	0.835 (0.833, 0.836)	0.952 (0.950, 0.953)	0.962 (0.961, 0.964)
College or more	0.479 (0.478, 0.480)	0.647 (0.646, 0.649)	0.655 (0.653, 0.656)
**Household income (reference, 0–130% FPL)**			
>130–350% FPL	0.407 (0.406, 0.407)	0.456 (0.456, 0.457)	0.456 (0.455, 0.457)
>350% FPL	0.230 (0.230, 0.231)	0.274 (0.273, 0.274)	0.277 (0.276, 0.277)
**HEI status (reference, Inadequate)**			
Average	0.766 (0.765, 0.767)	0.842 (0.841, 0.843)	0.848 (0.846, 0.849)
Optimal	0.378 (0.377, 0.379)	0.448 (0.447, 0.450)	0.455 (0.453, 0.456)
**BMI (reference, Normal or low weight)**			
Overweight		0.844 (0.843, 0.846)	0.833 (0.832, 0.835)
Obese		1.341 (1.339, 1.343)	1.265 (1.263, 1.267)
**Smoking status (reference, Never)**			
Former		1.286 (1.284, 1.288)	1.274 (1.272, 1.276)
Current		2.546 (2.542, 2.550)	2.564 (2.560, 2.568)
**Drinking status (reference, None)**			
Light		0.955 (0.954, 0.957)	0.971 (0.969, 0.973)
Moderate		0.702 (0.701, 0.703)	0.722 (0.720, 0.723)
Heavy		1.141 (1.139, 1.144)	1.181 (1.179, 1.184)
**Diabetes (reference, No)**			
YES			1.637 (1.634, 1.640)

FPL = family income to poverty; CI = confidence interval; OR = odds ratio. Model 1 = adjusted for demographics characteristics (i.e., sex, age group, race, income, and education); Model 2 = Model 1 covariates + BMI, smoking, and drinking status; Model 3 = Model 2 covariates + diabetes.

## Data Availability

Publicly available datasets were analyzed in this study. This data can be found here: [https://wwwn.cdc.gov/nchs/nhanes/ContinuousNhanes/Default.aspx?BeginYear=2005].

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
