# Peer review of "Higher HEI-2015 Score Is Associated with Reduced Risk of Depression: Result from NHANES 2005–2016"

_nutrients, 2021, doi:10.3390/nu13020348_

Round 1

Reviewer 1 Report

This is important and valuable article. BUT I've indicated some minor addition to the text which might improve it.

Both the method and the presentation of results are appropriate. The discussion is well conducted and what is more it includes some sound arguments concerning practical application of results.
However I would suggest some comments on the limitations of the study to be added to the discussion section. It would be worthwhile to propose some further research on what direction they could go in?

Author Response

Response to Reviewer 1 Comments

Point 1: This is important and valuable article. BUT I've indicated some minor addition to the text which might improve it.

Both the method and the presentation of results are appropriate. The discussion is well conducted and what is more it includes some sound arguments concerning practical application of results.

However I would suggest some comments on the limitations of the study to be added to the discussion section. It would be worthwhile to propose some further research on what direction they could go in? 

Response 1: Thank you for your valuable advice. In the end of the discussion, we have included three paragraphs. The first paragraph is about the limitations and strengths of our design. The second paragraph is about our plan to figure out the mechanism how HEI score influences depression. The final paragraph is about practical applications on the basis of your suggestions. (Line 261-274 Page 13 in “Discussion” section of the revised manuscript)

Reviewer 2 Report

Introduction. Dear Authors. Write all the paragraphs in a single paragraph. Please rewrite lines 32 and 33. I understand what you mean, but it is not aesthetic to use scripts. Add a small paragraph about the incidence of depression in diferents ages of the life. The introduction is brief, but clear and precise.

Materials and method: adapt figure 1 to the Prism model for theoretical reviews and meta-analysis.
Measures. Remove the dashes. Add reliability data in the scaled questionnaires. Add in each questionnaire for which sample (nationality) has been scaled. The term gender can give rise to problems since it is a social construction, better use sex. At the level of education, use more generic terms such as university or doctorate, which are used in all countries. In the "smoking status" questionnaire, add the number of items, how they are valued, scale, etc., as you did in the previous ones. Remember to add also number of items in "drinking status".

Statistical analysis. Specify the method of regression used: steps forward, backward, etc. And the why of your choice.

Results:

The titles of the figures should be somewhat more concrete.

The results are interesting but need to be written up more. In the same way that you have explained the results of table 3, do the same with the rest of the tables and figures. The development of table 1 is sufficient. It is necessary to expand the rest.

Discussion. It is necessary that all the references in the theoretical framework appear in the discussion. More studies are needed to make a comparison and deep analysis. Add at the end of the discussion a paragraph on limitations, another on prospective studies and another on practical applications.

Author Response

Response to Reviewer 2 Comments

Point 1: Introduction. Dear Authors. Write all the paragraphs in a single paragraph. Please rewrite lines 32 and 33. I understand what you mean, but it is not aesthetic to use scripts. Add a small paragraph about the incidence of depression in different ages of the life. The introduction is brief, but clear and precise.

Response 1: Thank you for your valuable comments. We have merged all the paragraphs into a single paragraph. Line 32 and 33 have been rewritten according to your valuable advice. To expand the introduction, we also add a paragraph about the incidence of depression in different age groups. (Line 32-34, 37-39 Page 1 in “Introduction” section of the revised manuscript)

Point 2: adapt figure 1 to the Prism model for theoretical reviews and meta-analysis.

Response 2: Thank you for your comments. Figure 1 is adapted to the Prisma model and presented below.

Point 3: Measures. Remove the dashes.

Response 3: Thank you for your valuable comments. The dashes are removed according to your valuable comments. (Line 75,83,91 Page 4 in “Measures” section of the revised manuscript)

Point 4: Add reliability data in the scaled questionnaires. Add in each questionnaire for which sample (nationality) has been scaled.

Response 4: Thank you for your valuable comments. I will answer the question as I comprehend.

The reliability data of PHQ-9 questionnaire was added in the paragraph as Cronbach’s α = .89.

Weights are created in NHANES to account for the complex survey design (including oversampling), survey non-response, and post-stratification adjustment to match total population counts from the Census Bureau. When a sample is weighted in NHANES it is representative of the U.S. civilian noninstitutionalized resident population. A sample weight is assigned to each sample person. It is a measure of the number of people in the population represented by that sample person.

And we adopt Fasting Subsample 2 Year MEC Weight since we must use the weight of the smallest subpopulation that includes all the variables I want to include in my analyses.

When combining two or more two-year cycles from 2001–2002 onward, new multi-year sample weights can be computed by simply dividing the two-year sample weights by the number of two-year cycles in the analysis.

We utilize data from six circles in 2005-2016, so sample weight in our analyses should be Fasting Subsample 2 Year MEC Weight divided by six.

Point 5: The term gender can give rise to problems since it is a social construction, better use sex.

Response 5: Thank you for your valuable advice, the term gender is modified into sex now. (Line 94,152,161,166,178,180 of the revised manuscript)

Point 6: At the level of education, use more generic terms such as university or doctorate, which are used in all countries.

Response 6: Thank you for your valuable advice.

In our analyses, the level of education was categorized as less than a high school diploma (reference group), high school graduate/GED, some college/AA degree, and college graduate or more according to the questionnaire below. For instance, the term college graduate or more include bachelor, master and doctorate, so we did not modify the description of the questionnaire into more generic terms. Hope for your understanding.

Point 7: In the "smoking status" questionnaire, add the number of items, how they are valued, scale, etc., as you did in the previous ones. Remember to add also number of items in "drinking status".

Response 7: Thank you for your valuable comments.

We add the items that respondent was questioned and adjust the order of the sentences, so that the scale of smoking status and drinking status can be found at a glance.

The scale of smoking status was common among NHANES articles, such as “Cigarette Smoking in Persons Living with Hepatitis C: The National Health and Nutrition Examination Survey (NHANES), 1999-2014” and “Blood cadmium by race/Hispanic origin: The role of smoking”.

The scale of drinking status varies among NHANES articles. In the article entitled “Association of Hepatitis C Virus With Alcohol Use Among U.S. Adults: NHANES 2003–2010”, lifetime abstainers reported <12 drinks ever. Former drinkers reported ≥12 drinks in their lifetime but none in the past year. Non-excessive current drinkers, on average, reported ≤14 drinks/week for men or, on average, seven or fewer drinks/week for women, and never five or more drinks/day in a single day during the past year for either. Excessive current drinkers, on average, reported >14 drinks/week for men or, on average, more than seven drinks/week for women, or five or more drinks/day in a single day at least once during the past year for either. Another article entitled “Dietary Patterns and Alcohol Consumption During Pregnancy: Secondary Analysis of Avon Longitudinal Study of Parents and Children” stratified the extent of alcohol consumption in subjects who reported a history of alcohol use into four groups using quartiles. And in our analyses, drinking status was categorized into four strata (0, < 1, 1– < 8, and ≥8 drinks per week) and defined as none (reference group), light, moderate, and heavy alcohol consumption, respectively. (Line 105-107,112-119 Page 5 in “Measures” section of the revised manuscript)

Point 8: Statistical analysis. Specify the method of regression used: steps forward, backward, etc. And the why of your choice.

Response 8: Thank you for your valuable comments. We adopt steps forward (likelihood ratio) binary logistic regressions in our analyses. Since our dependent variable is categorized into no depression and with depression, binary logistic regression is suitable for our analyses.

The likelihood ratio test (LRT) is a statistical test of the goodness-of-fit between models with different independent variables. It is not only suitable for hypothesis test of single independent variable, but also suitable for simultaneous test of multiple independent variables.

(Line 130 Page 6 in “Statistical Analysis” section of the revised manuscript)

Point 9: Results: The titles of the figures should be somewhat more concrete.

Response 9: Thank you for your valuable advice.

The title of Figure 1 is modified into “Flowchart of the study population”. (Line 71 Page 3 in “Sample” section of the revised manuscript)

The title of Figure 2 is modified into “The trends of HEI-2015 score in the six circles from 2005-2016 in NHANES”. (Line 190-191 Page 10 in “Results” section of the revised manuscript)

The title of Figure 3 is modified into “The forest plot shows the odds ratios of analyzing variables in 3 weighted logistic regression models”. (Line 199-201 Page 11 in “Results” section of the revised manuscript)

Point 10: The results are interesting but need to be written up more. In the same way that you have explained the results of table 3, do the same with the rest of the tables and figures. The development of table 1 is sufficient. It is necessary to expand the rest.

Response 10: Thank you for your valuable comments. As you can see, the results of several figures and tables are expanded like Table 3. We add several descriptions about baseline characteristics of the final study sample and deep information in Table 1. (Line 136, 140-141, 145-150, 185-188, 194-197 in “Results” section of the revised manuscript)

Point 11: Discussion. It is necessary that all the references in the theoretical framework appear in the discussion. More studies are needed to make a comparison and deep analysis.

Response 11: Thank you for your valuable comments. All results about variables in the theoretical framework are explained. In the discussion, we add plenty of new articles for comparison. (Line 218-227 Page 12 in “Discussion” section of the revised manuscript)

Point 12: Add at the end of the discussion a paragraph on limitations, another on prospective studies and another on practical applications.

Response 12: Thank you for your valuable comments. Three paragraphs are included to enrich the article according to your key and prospective comments. (Line 261-274 Page 13 in “Discussion” section of the revised manuscript)

Reviewer 3 Report

Referee Report

MDPI Nutrients

nutrients-1060962

Higher HEI-2015 score is associated with reduced

risk of depression: result from NHANES 2005-2016

Authors applied logistic regression and Chi-square test to examine risk factors

for depression using the NHANES data set between 2005 and 2016. Software SPSS

(version 23.0) was utilized for the univariate analyses and multivariate analyses. This

research paper is well-organized. A few comments are listed below.

Comments

_ Authors should briefly explain the trend test of temporal association they used

for Table 1. There are several parametric and nonparametric ways to test the

trend. Authors may justify their choice.

_ Authors presented three logistic regression models to show the relations between

depression and three different sets of covariates which all include HEI. Confidence

intervals were reported in Table 3. Since three models were considered

simultaneously, the authors should consider the issue of multiplicity. Besides

unadjusted confidence intervals, authors should also include the adjusted confidence

intervals for multiplicity.

_ Authors took 10349 samples from a total of 60936 records. There are 29306

patients whose ages are older than 20 and have complete information about

depression. Authors remove 18006 records because of missing data on any of the

covariates. Authors should include an analysis to partially show the similarity

between the 10349 samples used in this research and 18006 records removed

because of missing values.

_ Scott and Rao’s original paper should be cited.

1

_ Abbreviation HEI should be explained when it first appears in abstract and

introduction: Healthy Eating Index (HEI).

_ Figure 3 is a little bit difficult to read. Authors may separate the numbers and

plots in Figure 3, and use more distinct colors for Model 1 and Model 3.

_ Minor: RACE in Table 3 should be Race.

2

Author Response

Response to Reviewer 3 Comments

Point 1: Authors should briefly explain the trend test of temporal association they used for Table 1. There are several parametric and nonparametric ways to test the trend. Authors may justify their choice.

Response 1: Thank you for your valuable suggestions. There are indeed several methods to test the trend, such as Cochran-Armitage trend test, Manner-Kendall trend test and orthogonal polynomial contrasts. For our analyses, we adopted Cochran-Armitage trend test in Table 1 for temporal association. In the test, we set data in six cycles as an ordinal categorical variable and depression as a binary variable, which met the requirements of Cochran-Armitage trend test. We also consider treating the 2-year survey cycle as a continuous variable in survey-weighted linear regression models, but we have a belief that it may destroy the structure of the article. And if you recommend other methods, please inform us to conduct analysis. (Line 128 Page 5 in “Statistical Analysis” section of the revised manuscript)

Point 2: Authors presented three logistic regression models to show the relations between depression and three different sets of covariates which all include HEI. Confidence intervals were reported in Table 3. Since three models were considered simultaneously, the authors should consider the issue of multiplicity.

Response 2: Thank you for your valuable comments. Maybe you mean multi-collinearity, and we will answer the question according to this.

    We conduct a collinearity diagnose and the result is presented below. Since the tolerance are all greater than 0.1 and VIF are all less than 5, we have a belief that severe multi-collinearity does not exist in these models.

Point 3: Besides unadjusted confidence intervals, authors should also include the adjusted confidence intervals for multiplicity.

Response 3: Thank you for your valuable comments.

In table 2, we conducted chi-square tests and P values for all variables are less than 0.001, so all the variables can be included in the models.

Model 1 was adjusted for demographics characteristics (i.e., sex, age group, race, income, and education). Model 2 was adjusted for all Model 1 covariates and BMI, smoking, and drinking status. Moreover, Model 3 was adjusted for all Model 2 covariates and diabetes.

All odds ratio appeared in the models were adjusted odd ratios. Sorry for our mistake that we have not explained these models clearly.

Point 4: Authors took 10349 samples from a total of 60936 records. There are 29306 patients whose ages are older than 20 and have complete information about depression. Authors remove 18006 records because of missing data on any of the covariates. Authors should include an analysis to partially show the similarity between the 10349 samples used in this research and 18006 records removed because of missing values.

Response 4: Thank you for your valuable comments. It is our mistake that we have not conducted such analysis about baseline characteristics between the 10349 samples used in this research and 18006 records removed because of missing values.

We adopted Fasting Subsample 2 Year MEC Weight in our analysis which is not owned by all participants with complete information about depression. Thus, we cannot conduct the analysis with Fasting Subsample 2 Year MEC Weight. However, our analyse are all based on Fasting Subsample 2 Year MEC Weight. If we compare these two groups with other weights like Dietary Day One Sample Weight, all our analyses will ruin for the use of multiple weights. Hope for your understanding.

Point 5: Scott and Rao’s original paper should be cited.

Response 5: Thank you for your valuable comments and the paper was cited when Scott-Rao chi-square test was mentioned. (Line 373-374 Page 15 in “References” section of the revised manuscript)

Point 6: Abbreviation HEI should be explained when it first appears in abstract and introduction: Healthy Eating Index (HEI).

Response 6: Sorry for our careless behaviour, abbreviation HEI is now explained when it first appears in abstract and introduction. (Line 12 Page 1 in “Abstract” section and line 52 Page 2 in “Introduction” section of the revised manuscript)

Point 7: Figure 3 is a little bit difficult to read. Authors may separate the numbers and plots in Figure 3, and use more distinct colours for Model 1 and Model 3.

Response 7: Thank you for your valuable advice and we redraw the figure in which the numbers are eliminated. The new figure is presented below and the numbers are in Table 3. Besides, we endow Model 1 gold colour and Model 3 purple colour for obvious comparison.

Point 8: Minor: RACE in Table 3 should be Race.

Response 8: Sorry for our misspelling, and RACE in Table 3 is modified into Race. (Line 178 Page 9 in “Result” section of the revised manuscript)

Round 2

Reviewer 2 Report

Dear authors,
The manuscript has improved significantly with the modifications made. My congratulations. The introduction is correct, the methodology clear and concentrated. The results are well explained and the discussion and conclusion are consistent with the results.